# Retinal Vascular Events after mRNA and Adenoviral-Vectored COVID-19 Vaccines—A Case Series

**DOI:** 10.3390/vaccines9111349

**Published:** 2021-11-17

**Authors:** Christian Girbardt, Catharina Busch, Mayss Al-Sheikh, Jeanne Martine Gunzinger, Alessandro Invernizzi, Alba Xhepa, Jan Darius Unterlauft, Matus Rehak

**Affiliations:** 1Department of Ophthalmology, University Hospital Leipzig Medical Center, 04103 Leipzig, Germany; christian.girbardt@medizin.uni-leipzig.de (C.G.); J.D.Unterlauft@gmx.de (J.D.U.); matus.rehak@medizin.uni-leipzig.de (M.R.); 2Department of Ophthalmology, University Hospital Zurich, 8091 Zurich, Switzerland; mayss.alsheikh@gmail.com (M.A.-S.); Jeanne.gunzinger@usz.ch (J.M.G.); 3Eye Clinic, Department of Biomedical and Clinical Sciences “Luigi Sacco”, Luigi Sacco Hospital, University of Milan, 20157 Milan, Italy; alessandro.invernizzi@gmail.com (A.I.); xhepaalba@yahoo.com (A.X.); 4Save Sight Institute, Discipline of Ophthalmology, Sydney Medical School, The University of Sydney, Sydney, NSW 2000, Australia; 5Department of Ophthalmology, University Hospital Bern, Inselspital, 3010 Bern, Switzerland

**Keywords:** post-vaccination, COVID-19 vaccination, vascular complication, retinal vascular complication

## Abstract

Background: To describe cases of retinal vascular events shortly after administration of mRNA or adenoviral-vectored COVID-19 vaccines. Design: Retrospective, multicenter case series. Methods: Six cases of retinal vascular events shortly after receiving COVID-19 vaccines. Results: A 38-year-old, otherwise healthy male patient presented with branch retinal arterial occlusion four days after receiving his second dose of SARS-CoV-2 vaccination with Comirnaty^®^ (BioNTech^®^, Mainz, Germany; Pfizer^®^, New York City, NY, USA). An 81-year-old female patient developed visual symptoms twelve days after the second dose of SARS-CoV-2 vaccination with Comirnaty^®^ and was diagnosed with a combined arterial and venous occlusion in her right eye. A 40-year-old male patient noticed blurry vision five days after his first dose of SARS-CoV-2 vaccination with Comirnaty^®^ and was diagnosed with venous stasis retinopathy in his left eye. A 67-year-old male was diagnosed with non-arteritic anterior ischemic optic neuropathy in his right eye four days after receiving the first dose of Vaxzevria^®^ (AstraZeneca^®^, Cambridge, UK). A 32-year-old man presented with a sudden onset of a scotoma two days after receiving the second dose of SARS-CoV-2 vaccination with Spikevax^®^ (Moderna, Cambridge, UK) and was diagnosed with a circumscribed nerve fiber infarction. A 21-year-old female patient developed an acute bilateral acute macular neuroretinopathy three days after receiving the first dose of SARS-CoV2-vaccine Vaxzevria^®^ (AstraZeneca^®^, Cambridge, UK). Conclusion: This case series describes six cases of retinal vascular events shortly after receiving mRNA or adenoviral-vectored COVID-19 vaccines. The short time span between received vaccination and occurrence of the observed retinal vascular events raises the question of a direct correlation. Our case series adds to further reports of possible side effects with potential serious post-immunization complications of COVID-19 vaccinations.

## 1. Introduction

After efficacy was reported in clinical trials, several COVID-19 vaccines received emergency authorizations for their use in a large number of countries. Among others, messenger ribonucleic acid (mRNA)-based vaccines and viral-vectored vaccines have been approved for use [1]. RNA vaccines encode the stabilized prefusion form of the SARS-CoV-2 spike glycoprotein trimer and offer the advantages of being non-infectious, not integrating into the host genome, and not inducing vector-specific responses [1]. Viral-vectored vaccines, on the other hand, employ recombinant viruses modified to encode antigens derived from the target pathogen to directly infect host cells [1]. Vaccines elicit immunological mechanisms to induce protection against a certain disease. Globally, research found that the efficacy of all vaccines exceeded 70%, and RNA-based vaccines had the highest efficacy over 94% [2]. However, vaccines also bear the risk of immune-mediated adverse reactions. In times of the COVID-19 pandemic, the safety of COVID-19 vaccines is of special interest. In the past few months, an increasing number of cases of suspected vaccine-related arterial and venous thrombotic events after mRNA-based and adenoviral-vectored COVID-19 vaccines were published [3]. Here, we report a case series of retinal events shortly after vaccination with the mRNA vaccine Comirnaty^®^ (BioNTech^®^, Mainz, Germany; Pfizer^®^, New York City, NY, USA), the mRNA vaccine Spikevax^®^ (Moderna^®^, Cambridge, UK), and the adenoviral-vectored vaccine ChAdOx1 nCoV-19 (University of Oxford, UK/AstraZeneca^®^, Cambridge, UK).

### 1.1. Case 1: Branch Retinal Artery Occlusion 

A 38-year-old male patient presented with a two-day old painless visual field loss of the inferior hemisphere in his right eye (Table 1). He had received his second dose of the SARS-CoV-2 vaccination with Comirnaty^®^ (BioNTech^®^, Mainz, Germany; Pfizer^®^, New York City, NY, USA) three days before the event.

We saw a healthy athletic patient who did not have concomitant diseases apart from an ankylosing spondylitis, which had not required any therapy for three years. Best-corrected visual acuity (BCVA) was 1.0 (decimal) in both eyes. There was no relative afferent pupillary defect. Automated-computed perimetry confirmed an inferior visual field defect in the right eye. Ophthalmoscopy revealed a marked retinal edema around the superior temporal arcade reaching the optic disc and the macula (Figure 1A). Optical coherence tomography angiography (OCTA) showed missing retinal blood supply in this area (Figure 1B). Ophthalmoscopy of the left eye was normal.

We diagnosed a branch retinal artery occlusion in the right eye. Laboratory work-up was normal except for a borderline value of low-density lipoprotein (1.84 mg/dl). Specific thrombophilia diagnostic workup detected no abnormalities. Erythrocyte sedimentation rate was 31/59 mm, which we linked to the known ankylosing spondylitis. Brain magnetic resonance imaging (MRI) was normal for his age. Duplex ultrasonography confirmed flow reduction of the central retinal artery of the right eye. Apart from this, no relevant stenoses or occlusions of the extra- and intracranial vessels were found. Lumbar puncture detected no abnormalities. Transesophageal echocardiography showed a patent foramen ovale. Deep vein thrombosis was excluded via ultrasonography. We started platelet aggregation inhibition with acetylsalicylic acid and a lipid-lowering medication with simvastatin. Interventional patent foramen ovale closure was waived.

### 1.2. Case 2: Bilateral (Combined) Retinal Occlusion

An 81-year-old female patient presented herself in the emergency room with unsteadiness and blurred vision in her right eye, which had increased over the last couple of days (Table 1). The patient had received the second dose of the SARS-CoV-2 vaccination with Comirnaty^®^ twelve days before the onset of visual symptoms. She suffered from arterial hypertension, which was well-controlled by oral medication. At first presentation, BCVA was 0.05 (decimal) in the right eye and 0.25 (decimal) in the left eye. Ophthalmological examination revealed multiple intraretinal hemorrhages in the right eye and centrally localized hemorrhages and hard exudates in the left eye (Figure 2A,B). The macula of the right eye was intact, and the macula of the left eye showed an intraretinal edema with exudates. A secondary finding was an enlarged cup-disc-ratio more pronounced in the left eye, which was due to a known primary open-angle glaucoma. Fundus fluorescein angiography (FFA) revealed arterial capillary non-perfusion and a delayed venous filling in the right eye and vessel leakage inferior to the fovea in the left eye (Figure 2C,D). Optical coherence tomography (OCT) showed a hyperreflectivity of the inner retinal layers in the right eye and a cystoid macular edema in the left eye (Figure 2E,F). 

Neurological work-up remained without abnormality. Cerebral sinus venous thrombosis could be ruled out via computed tomography angiography (CTA).

We diagnosed a combined arterial and venous occlusion of the right eye and a (most likely pre-existing) branch retinal vein occlusion of the left eye. We initiated intravitreal anti-vascular endothelial growth factor (anti-VEGF) therapy in the left eye.

### 1.3. Case 3: Venous Stasis Retinopathy

A 40-year-old male patient presented with blurry vision and perception of a central grayish spot in his left eye (Table 1). He had his first dose of the SARS-CoV-2 vaccination with Comirnaty^®^ (BioNTech^®^, Mainz, Germany; Pfizer^®^, New York City, NY, USA) five days before this event. 

We saw an otherwise healthy patient without any concomitant disease. Clinical examination did not reveal any neurological symptoms.

BCVA was 1.25 (decimal) in the right eye and 0.8 (decimal) in the left eye at initial presentation. Slit-lamp examination of the anterior segments was within normal limits in both eyes. Ophthalmoscopy revealed no pathological findings in the right eye (Figure 3A) but showed intraretinal bleedings and vascular tortuosity of the inferior retinal hemisphere and a parapapillary cotton wool spot in the left eye (Figure 3B). OCT of the macula was within normal limits in the right eye (Figure 3C). OCT of the left eye showed a slight swelling of the inner retina in the papillo-macular region (Figure 3D). 

We diagnosed venous stasis retinopathy without significant macular edema in the left eye.

Control examination after two weeks revealed regression of the cotton wool spot but persisting intraretinal bleedings. Visual acuity of the left eye increased to 1.25 (decimal). Wide-angle FFA at this time was within normal limits in the right eye (Figure 3E) and showed blocked fluorescence, corresponding to intraretinal hemorrhages, but no ischemic lesions in the left eye (Figure 3F).

### 1.4. Case 4: Non-Arteritic Anterior Ischemic Optic Neuropathy

A 67-year-old male presented to our ophthalmological emergency service with decreased vision and scotomata in his right eye, which he had been experiencing for two days (Table 1). He had received the first dose of Vaxzevria^®^ (AstraZeneca^®^, Cambridge, UK) four days prior to presentation (two days before the onset of ocular symptoms). After the vaccination, he had suffered from headaches and a high temperature (39 °C) for two days, which were treated with paracetamol (1000 mg 3 times/day). He suffered from diabetes and hypercholesterinemia, which were both well-controlled with no signs of diabetic retinopathy at his last funduscopic visit six months before.

On ophthalmological examination, BCVA was 0.1 (decimal) in the right eye and 1.0 (decimal) in the left eye. Anterior segment was within normal limits in both eyes. Intraocular pressure was 14 mmHg in the right eye and 16 mmHg in the left eye. Funduscopic examination of the right eye revealed an elevated and congested optic nerve head with surrounding intraretinal hemorrhages and cotton wool spots, especially along the superior margins of the disk (Figure 4A). The papillo-macular bundle appeared abnormal suggesting the presence of intraretinal fluid (Figure 4A). The left fundus was without pathological findings (Figure 4B). Both eyes showed no signs of diabetic retinopathy. FFA revealed staining at the optic nerve head (Figure 4C) in the right eye and was unremarkable in the left eye (Figure 4D). Indocyanine green angiography (ICGA) showed unremarkable findings in the choroid in both eyes. OCT of the right eye showed an elevated disk and intraretinal fluid along the papillo-macular bundle, suggesting a possible diagnosis of anterior ischemic optic neuropathy. The neurological examination was within normal limits. Full blood count, CPR, erythrocyte sedimentation rate, PT, PTT, fibrinogen, and D-Dimer were within normal limits. Brain CT and CTA scans revealed no abnormalities.

The patient was diagnosed with non-arteritic anterior ischemic optic neuropathy (NAION) in the right eye. The patient was scheduled for repetition of blood and coagulation tests and a follow-up at our neuro-ophthalmological service a week later.

### 1.5. Case 5: Nerve Fiber Infarction

A 32-year-old man presented with a sudden onset of a scotoma in his right eye. His prior ophthalmic history was unremarkable (Table 1). An attention deficit hyperactivity disorder was known, which was treated with methylphenidate for more than ten years. Cardiovascular risk factors and a history of other drugs intake were denied. Two days prior to the symptom’s onset, he had received the second dose of the SARS-CoV-2 vaccination Spikevax^®^ (Moderna, Cambridge, UK). 

His BCVA was 1.0 (decimal) in both eyes. Intraocular pressure was within normal limits. The slit-lamp examination of the anterior segment was normal in both eyes. Dilated fundus examination revealed a greyish-whitish spot with fluffy margins temporal to the optic disc in the right eye (Figure 5A) and no pathological findings in the left eye (Figure 5B).

The wide-field FFA showed normal filling and transit time. A focal area of hypofluorescence temporal to the optic disc was visible on the FFA in the early phase of the right eye (Figure 5C). In the late phase, a decent hyperfluorescence defect perpendicular to the optic disc margin was detectable. The FFA was normal otherwise in the right eye as well as the left eye (Figure 5D). The ICGA was unremarkable. The optical coherence tomography of the macula of the right eye showed thickening confined to the retinal nerve fiber layer (Figure 5E) and was normal in the left eye (Figure 5F). 

The patient was diagnosed with a cotton-wool spot, as a sign of a nerve fiber layer infarction, in the right eye. 

### 1.6. Case 6: Bilateral Acute Macular Neuroretinopathy

A 21-year-old female patient presented to our ophthalmological emergency service because of a circumscribed scotoma in her left eye (Table 1). She had received the first dose of the SARS-CoV2-vaccine Vaxzevria^®^ (AstraZeneca^®^, Cambridge, UK) three days before. She had no pre-existing medical conditions and was not taking any medication. 

BCVA was 1.0 (decimal) in both eyes. Automated-computed perimetry showed small, round, centrally located circumscribed scotomas in both eyes. Slit-lamp examination of the anterior segments and fundoscopic examination were unremarkable. Infrared images (Figure 6A,B) and en-face OCT images of the level of the ellipsoid zone (Figure 6C,D) showed parafoveal lesions in both eyes. B-scan OCT revealed corresponding subtle alterations of the outer retinal layers, more pronounced in the left eye, that were not detectable upon ophthalmoscopic examination (Figure 6E,F). OCTA showed normal perfusion in both eyes. Neurological examination did not reveal any pathological findings. Magnetic resonance angiography (MRA) ruled out signs of sinus venous thrombosis or cerebral vasculitis. Connective tissue disease and vasculitis screening showed normal values except for slightly elevated proteinase 3 antibodies. 

We diagnosed bilateral acute macular neuroretinopathy (AMN). 

## 2. Discussion

Apart from pulmonary symptoms, SARS-CoV2 infection causes thromboembolic complications through direct endothelial damage, as well as through the secondary immune reaction [4,5]. Thromboembolic events have also been described after SARS-CoV2 vaccinations with vector-based agents and might also appear, to a minor degree, after vaccination with mRNA-based agents [3]. Here, we report a case series of different retinal vascular events shortly after administration of mRNA and adenoviral-vectored COVID-19 vaccines.

Cases 1 and 2 show retinal vascular occlusions, and Case 3 shows a venous stasis retinopathy, which is beleved to be a preliminary stage of a clinically detectable retinal venous occlusion. Retinal vein occlusions have a global prevalence of 5 per 1000 [6]. Retinal artery occlusions show an estimated prevalence of 1–5 per 100,000 and can be seen as comparable to an ischemic stroke, leading to the recommendation of a diagnostic work-up [7,8,9]. Risk factors for retinal vascular occlusion include age, smoking, hypertension, arteriosclerosis, diabetes mellitus, hyperlipidemia, blood hyperviscosity, and thrombophilia [10]. Cardiac pathologies are important causes of retinal artery occlusions, especially in young individuals [11]. The Case 1 patient had a patent foramen ovale. This finding does not have a higher risk for stroke development in the general population, but it can lead to stroke via paradoxical embolism when combined with venous clots. In a cryptogenic stroke, mechanical closure of a patient foramen ovale can reduce the risk for stroke relapses [12]. There are no specific recommendations concerning this topic in retinal artery occlusions. Thus, an occlusion of the patent foramen ovale was waived in the presented case but might have to be re-evaluated in the case of future SARS-CoV2 booster injections.

The Case 4 patient presented with a NAION. The pathophysiology of NAION is still being discussed and is controversial. However, it is presumed to result from an infarct within the retrolaminar portion of the optic nerve head. 

Case 5 showed a singular parapapillary cotton-wool spot. Cotton-wool spots are believed to represent nerve fiber layer infarctions due to ischemia from a retinal arteriole obstruction. They can have an ischemic etiology due to diabetes or hypertension or can occur due to inflammatory or immune conditions. However, the patient in our case was healthy without any known cardiovascular risk factors. 

Case 6 presented with bilateral AMN. Cases of AMN shortly after COVID-19 vaccination have been reported before [13]. Risk factors suggested to be associated with the development of AMN are preceded infection or febrile illness, female gender, and/or oral contraceptives [14]. Although the pathogenesis of AMN remains uncertain and is complex, a retinal microvascular etiology affecting the deep retinal capillary plexus (DCP) is suggested [14,15]. It is plausible that relative hypovolemia, associated with fever and flu-like illness after vaccination or eventually subclinical small retinal capillary vasculitis induced by vaccination itself, lead to an ischemia of the DCP. 

Arterial and venous thrombotic events have been reported especially after the use of viral-vectored vaccines [2,3]. However, whether thromboembolic events also occur in a higher frequency after the use of RNA vaccines is still under discussion, since some studies reported such events [16], while others could not detect a higher frequency of thromboembolic events afterward [2,17]. 

It remains unclear if the reported retinal vascular events in our study are specific side effects of COVID-19 vaccinations. Ocular side effects, such as retinal vascular occlusions, AMN, and anterior and posterior uveitis, were also observed after vaccination for other diseases, such as influenza or hepatitis B [15,18,19]. Reports on ocular side effects after COVID-19 vaccinations are rare so far [20]. Further multicenter longitudinal studies are required to analyze whether a direct association exists. 

## 3. Conclusions

In conclusion, COVID-19 vaccination is justified as an essential public health measure, and all authorized vaccines have been proven to be safe and effective. Our case series adds to other reports of possible side effects with potential serious post-immunization complications, which general physicians and ophthalmologist should be aware of.

## Figures and Tables

**Figure 1 vaccines-09-01349-f001:**
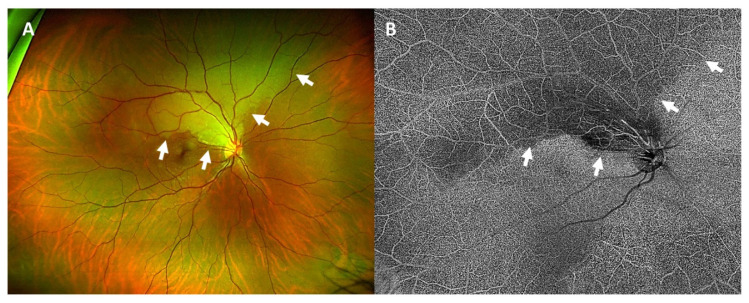
Wide-angle fundus photography of the right eye (**A**) showed a pale, edematous area corresponding to the area of arterial occlusion (white arrows). In optical coherence tomography angiography (**B**) the area of arterial occlusion was seen as a hypoautofluorescent area with a missing blood supply (white arrows).

**Figure 2 vaccines-09-01349-f002:**
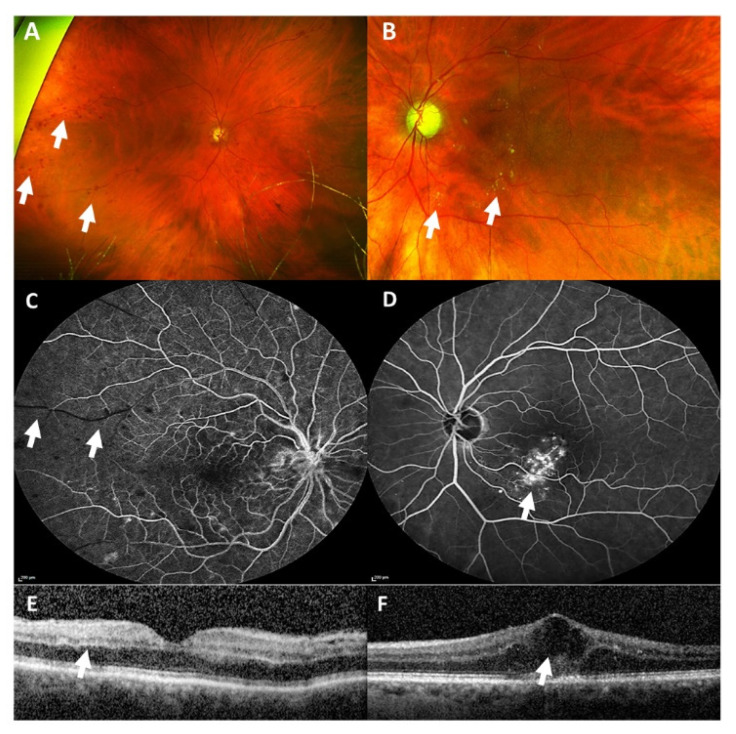
Wide-angle fundus photography (**A**,**B**) revealed multiple intraretinal hemorrhages in the right eye (white arrows, **A**) and hard exudates, seen as yellowish circumscribed spots, in the left eye (white arrows, **B**). Fluorescein angiography (**C**,**D**) showed an arterial capillary non-perfusion (white arrows, **C**) in the right eye and vessel leakage inferior of the macula in the left eye, seen as hyperfluorescent non-well circumscribed area (white arrow, **D**). Optical coherence tomography (**E**,**F**) revealed a hyperreflectivity of inner retinal layers as a sign of an arterial occlusion (white arrow, **E**) in the right eye, and a cystoid macular edema (white arrow, **F**) in the left eye.

**Figure 3 vaccines-09-01349-f003:**
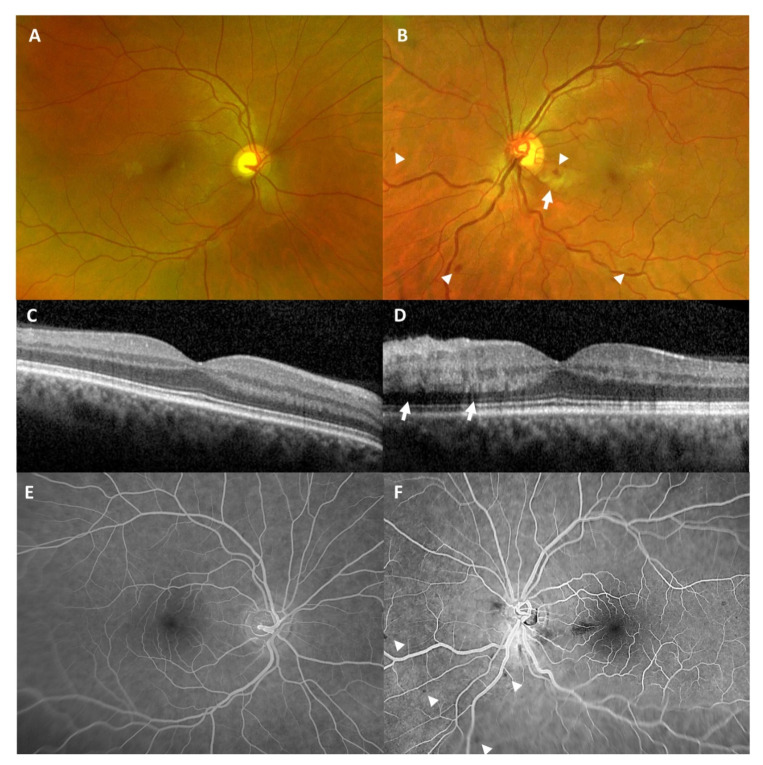
Wide-angle fundus photography (**A**,**B**) at initial presentation showed a normal right eye (**A**) and intraretinal hemorrhages (white arrow heads) as well as a parapapillary cotton-wool spot, seen as a fluffy yellow spot (white arrow), in the left eye (**B**). Optical coherence tomography of the macula (**C**,**D**) revealed a normal right eye (**C**) and a slight swelling of the inner retina (white arrows) in the left eye (**D**). Fluorescein angiography (**E**,**F**) two weeks later showed no abnormalities in right eye (**E**) and blocked fluorescence, due to persistent intraretinal hemorrhages (white arrow heads), but no ischemic lesions in the left eye (**F**).

**Figure 4 vaccines-09-01349-f004:**
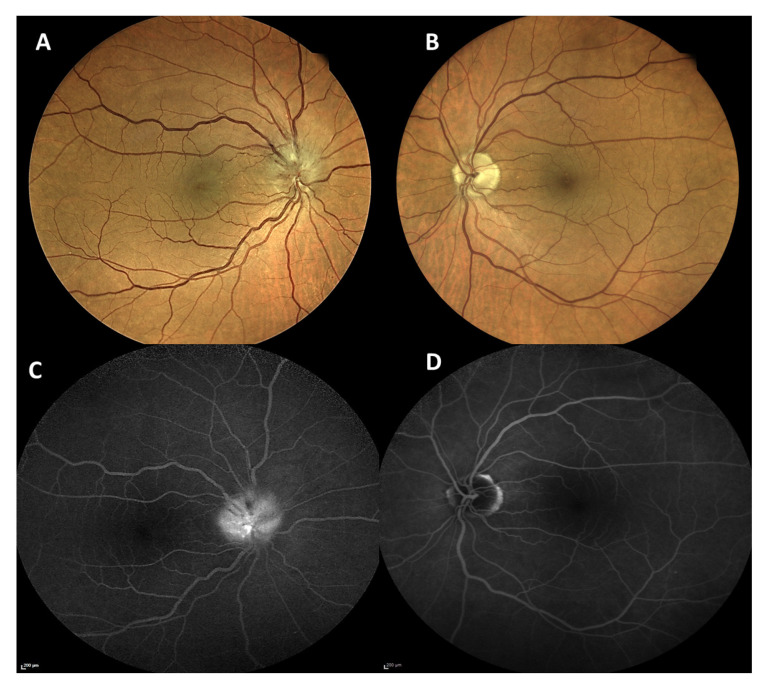
Fundus photography (**A**,**B**) showed an elevated and congested optic nerve head in the right eye (**A**) and was normal in the left eye (**B**). Late fundus fluorescein angiography (**C**,**D**) revealed a staining at the optic nerve head in the right eye (**C**) and was unremarkable in the left eye (**D**).

**Figure 5 vaccines-09-01349-f005:**
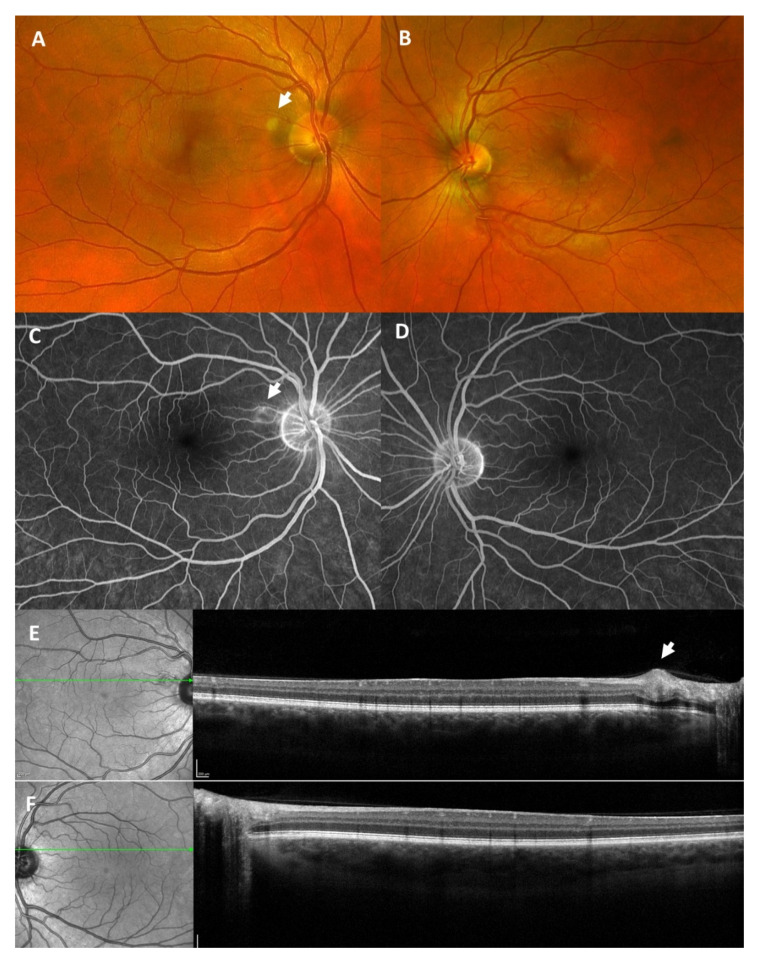
Fundus photography (**A**,**B**) revealed a cotton-wool spot temporal of the optic disc (white arrow) in the right eye (**A**) and was normal in the left eye (**B**). Fluorescein angiography (**C**,**D**) showed a hyperfluorescent spot (white arrow) corresponding to the cotton-wool spot in the right eye (**C**) and was normal in the left eye (**D**). Optical coherence tomography (**E**,**F**) revealed a circumscribed swelling of the retinal nerve fiber layer (white arrow) corresponding to the cotton-wool spot in the right eye (**E**) and was normal in the left eye (**F**). The green arrow indicates the position the OCT image was taken.

**Figure 6 vaccines-09-01349-f006:**
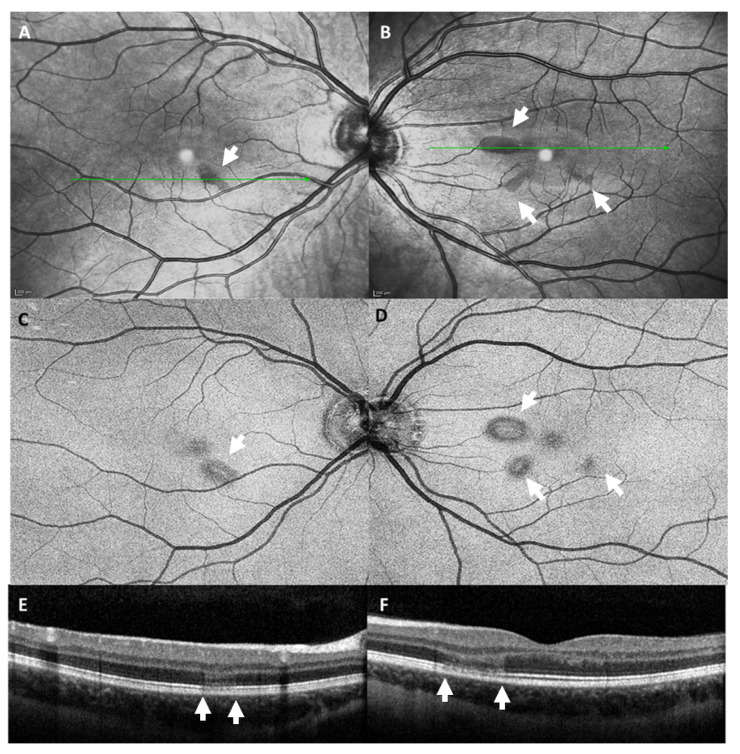
Infrared images (**A**,**B**) and en-face optical coherence tomography of the level of the ellipsoid zone (**C**,**D**) showed parafoveal lesions (white arrow) in both eyes. Optical coherence tomography of the macula (**E**,**F**) revealed subtle alterations of the outer retinal layers (area between the white arrows) in both eyes.

**Table 1 vaccines-09-01349-t001:** Summary of case characteristics.

Case	Age, Years	Gender	Comorbidities	Vaccine	Onset of Symptoms after Vaccination, Days	Symptoms	Diagnosis
#1	38	male	Ankylosing spondylitis, no therapy required	Comirnaty^®^, second dose	3	Painless visual field loss of inferior hemisphere right eye	Branch retinal artery occlusion
#2	81	female	Arterial hypertension, well controlled	Comirnaty^®^, second dose	12	Unsteadiness and blurred vision right eye	Combined arterial and venous occlusion
#3	40	male	None	Comirnaty^®^, first dose	5	Blurry vision, perception of greyish spot left eye	Venous stasis retinopathy
#4	67	male	Diabetes, Hypercholesterinemia, well controlled	Vaxzevria^®^, first dose	2	Decreased vision and scotoma right eye	Non-arteritic anterior ischemic optic neuropathy
#5	32	male	Attention deficit hyperactivity disorder, well controlled	Spikevax^®^, second dose	2	Scotoma right eye	Singular Cotton-wool spot
#6	21	female	None	Vaxzevria^®^, first dose	3	Scotoma left eye	Bilateral acute macular neuroretinopathy

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
