# Peer review of "Retinal Vascular Events after mRNA and Adenoviral-Vectored COVID-19 Vaccines—A Case Series"

_vaccines, 2021, doi:10.3390/vaccines9111349_

Round 1
Reviewer 1 Report
This manuscript by Girbardt et al. reported six cases of retinal vascular events shortly after getting different brand of COVID-19 vaccines, including the vaccines from Pfizer, Moderna and AstraZeneca. The figures presented by the authors are of high quality. I suggested the authors to have a deep discussion regarding the following questions:
- I am not sure these abnormal retinal vascular events are specific side effect of COVID-19 vaccine, it appears that the retinal vascular events are also reported in the patients after receiving Flu vaccine and other vaccines, I only listed several of these references in the following: (1) Williams, G. S., Evans, S., Yeo, D., & Al-bermani, A. (2015). Retinal artery vasculitis secondary to administration of influenza vaccine. BMJ case reports, 2015, bcr2015211971. https://doi.org/10.1136/bcr-2015-211971. (2) Liu JC, Nesper PL, Fawzi AA, Gill MK. Acute macular neuroretinopathy associated with influenza vaccination with decreased flow at the deep capillary plexus on OCT angiography. Am J Ophthalmol Case Rep. 2018 Feb 10;10:96-100. doi: 10.1016/j.ajoc.2018.02.008. PMID: 29541690; PMCID: PMC5849782. (3) Granel B, Disdier P, Devin F, Swiader L, Riss JM, Coupier L, Harlé JR, Jouglard J, Weiller PJ. Occlusion de la veine centrale de la rétine après vaccination contre l'hépatite virale B par vaccin recombinant. Quatre observations [Occlusion of the central retinal vein after vaccination against viral hepatitis B with recombinant vaccines. 4 cases]. Presse Med. 1997 Feb 1;26(2):62-5. French. PMID: 9082411.
- Are the retinal vascular events caused by the Adenoviral vector itself or due to the spike protein of the COVID-19? Some literature showed that AAV vectors may cause some immune response. (Bucher K, Rodríguez-Bocanegra E, Dauletbekov D, Fischer MD. Immune responses to retinal gene therapy using adeno-associated viral vectors - Implications for treatment success and safety. Prog Retin Eye Res. 2021 Jul;83:100915. doi: 10.1016/j.preteyeres.2020.100915. Epub 2020 Oct 15. PMID: 33069860.)
- What is the prevalence of these retinal vascular events in people before the COVID-19 pandemic? Are there any risk factors that associated with these retinal vascular events? With more people get vaccinated during the time of Covid-19 pandemic, is the prevalence of these retinal vascular events increased? For the prevalence of these retinal vascular events, is there any difference between vaccinated people and unvaccinated people? If the authors can find related reference, please give a brief discussion. Thanks for the invitation!
Author Response
This manuscript by Girbardt et al. reported six cases of retinal vascular events shortly after getting different brand of COVID-19 vaccines, including the vaccines from Pfizer, Moderna and AstraZeneca. The figures presented by the authors are of high quality. I suggested the authors to have a deep discussion regarding the following questions:
- I am not sure these abnormal retinal vascular events are specific side effect of COVID-19 vaccine, it appears that the retinal vascular events are also reported in the patients after receiving Flu vaccine and other vaccines, I only listed several of these references in the following: (1) Williams, G. S., Evans, S., Yeo, D., & Al-bermani, A. (2015). Retinal artery vasculitis secondary to administration of influenza vaccine. BMJ case reports, 2015, bcr2015211971. https://doi.org/10.1136/bcr-2015-211971. (2) Liu JC, Nesper PL, Fawzi AA, Gill MK. Acute macular neuroretinopathy associated with influenza vaccination with decreased flow at the deep capillary plexus on OCT angiography. Am J Ophthalmol Case Rep. 2018 Feb 10;10:96-100. doi: 10.1016/j.ajoc.2018.02.008. PMID: 29541690; PMCID: PMC5849782. (3) Granel B, Disdier P, Devin F, Swiader L, Riss JM, Coupier L, Harlé JR, Jouglard J, Weiller PJ. Occlusion de la veine centrale de la rétine après vaccination contre l'hépatite virale B par vaccin recombinant. Quatre observations [Occlusion of the central retinal vein after vaccination against viral hepatitis B with recombinant vaccines. 4 cases]. Presse Med. 1997 Feb 1;26(2):62-5. French. PMID: 9082411.
Response 1: We would like to thank the reviewer for raising this important point. We have added a discussion on retinal vascular events observed after other vaccinations to our manuscript (p.13, lines 288-293).
- Are the retinal vascular events caused by the Adenoviral vector itself or due to the spike protein of the COVID-19? Some literature showed that AAV vectors may cause some immune response. (Bucher K, Rodríguez-Bocanegra E, Dauletbekov D, Fischer MD. Immune responses to retinal gene therapy using adeno-associated viral vectors - Implications for treatment success and safety. Prog Retin Eye Res. 2021 Jul;83:100915. doi: 10.1016/j.preteyeres.2020.100915. Epub 2020 Oct 15. PMID: 33069860.)
Response 2: The reviewer raises a very interesting question whether vascular events might be caused by the AAV vector itself, and/or by the COVID-19 spike protein. To our best knowledge, the underlying events resulting in arterial and venous thrombotic events after COVID-19 vaccination are not fully understood yet and are still to be elucidated. After ocular delivery of AAV in terms of AAV-based gene therapy an induction of a local immune response was observed, potentially leading to retinal toxicity (Bucher K et al. Prog Retin Eye Res. 2021 Jul;83:100915.). However, whether this local immune response is directly from a host cell response to transgene expression or vector uptake or indirectly from a harmful immune response to the vector or the transgene product, or a combination of both, is still unclear. The fact that thrombotic events are also observed after non-vector-based vaccinations could support a possible immune response to the COVID-19 spike protein. However, since more evidence is needed to fully understand the pathophysiological events leading to vascular events after COVID-19 vaccination we decided to not further speculate on this in our discussion.
- What is the prevalence of these retinal vascular events in people before the COVID-19 pandemic? Are there any risk factors that associated with these retinal vascular events? With more people get vaccinated during the time of Covid-19 pandemic, is the prevalence of these retinal vascular events increased? For the prevalence of these retinal vascular events, is there any difference between vaccinated people and unvaccinated people? If the authors can find related reference, please give a brief discussion. Thanks for the invitation!
Response 3: The prevalence of the observed retinal vascular events differs highly between the different retinal events with a global prevalence of 5.2 per 1000 for any retinal vein occlusion (Laouri M, Chen E, Looman M, Gallagher M. The burden of disease of retinal vein occlusion: review of the literature. Eye (Lond). 2011;25(8):981-988. doi:10.1038/eye.2011.92Eye), and 1-5 per 100.000 for central arterial occlusion (Kido A et al. BMJ Open 2020;10:e041104. doi: 10.1136/bmjopen-2020-041104;; Leavitt JA et al. Am J Ophthalmol. 2011; 152:820–3.e2. doi: 10.1016/j.ajo.2011.05.005). AMN seems to be an even rarer event with the prevalence still being unclear. Risk factors for retinal vascular occlusion include age (as being the strongest risk factors), smoking, hypertension, arteriosclerosis, diabetes mellitus, hyperlipidemia, vascular cerebral stroke, blood hyperviscosity, and thrombophilia (Kolar P. J Ophthalmol. 2014;2014:724780. doi: 10.1155/2014/724780. Epub 2014 Jun 9. PMID: 25009743; PMCID: PMC4070325.). Risk factors being suggested to be associated with the development of AMN are preceded infection or febrile illness, female gender and/or oral contraceptives (Bhavsar KV et al. Surv Ophthalmol. 2016 Sep-Oct;61(5):538-65. doi: 10.1016/j.survophthal.2016.03.003.). To our best knowledge, there is no data on the exact incidence of these retinal vascular events in the vaccinated population. Thus, a direct comparison of incidences between vaccinated and unvaccinated cohorts is not possible yet and needs to be explored in the future. We have added information on the prevalence and risk factors for the observed retinal vascular events to the discussion (p.12 253-254 and 256-257; p.13 lines 276-277).
We would like the reviewer for the valuable comments.
Reviewer 2 Report
The article entitled ' Retinal vascular events after mRNA and adenoviral-vectored COVID-19 vaccines – A case series’ describes the side effects of administration of Covid-19 vaccination. The article showcases the side-effects in the retina that was observed upon administration.
- The article is well written, and the cases were well presented.
- The authors can include a summary table of the observation.
Patient age, sex, dosage regimen, manufacturer, Symptoms observed, diagnosis, treatment, patient recovery or followup details.
Author Response
Reviewer 2
The article entitled ' Retinal vascular events after mRNA and adenoviral-vectored COVID-19 vaccines – A case series’ describes the side effects of administration of Covid-19 vaccination. The article showcases the side-effects in the retina that was observed upon administration.
- The article is well written, and the cases were well presented.
- The authors can include a summary table of the observation.
Patient age, sex, dosage regimen, manufacturer, Symptoms observed, diagnosis, treatment, patient recovery or followup details.
Answer: We thank the reviewer for the helpful comments. We have added a table with the case characteristics accordingly (p.15).

Reviewer 3 Report
Girbardt et al. report the six cases of retinal vascular events shortly after administration of mRNA or adenoviral vector-based COVID-19 vaccines. The authors raise the questions of a direct correlation of the adverse responses of these cases against the COVID-19 vaccines as possible side effects with potential serious post-immunization complications of COVID-19 vaccinations. The report is timely and would be very important. However, healthy control images were not presented in some figures. And images were not understandable for non-experts without assisting explanations. Below are my specific comments.
- Introduction can cover more about mRNA and adenovirus vector vaccines in terms of efficacy and safety concerns, and how these previous reports could be related to the current report.
- Safety is often emphasized in mRNA vaccines over viral vectors, but the reported cases in this manuscript appear to show both types of vaccines, mRNA and adenovirus vectors, showed similar levels of safety concerns. It would be beneficial to discuss this point.
- Please add control to Fig. 1 so readers can easily understand the alteration.
- Line 67, missing retinal blood supply in this area. Where does this explain? Adding an arrow to the area would be helpful.
- In Fig. 2, are greed dots the hemorrhages and hard exudates in the left eye?
- In Fig. 2C and D, I understand the vessel leakage, but it is unclear about the description “FFA revealed arterial capillary non-perfusion and a delayed venous filling”.
- In Fig. 2, where were the OCT images taken? And please indicate the areas described as “a hyper-reflectivity of the inner retinal layers in the right eye and a cystoid macular edema”.
- In Fig. 3, please indicate the peri-papillary cotton wool spot. Since these images were taken by different imaging modalities, adding control images would be helpful to understand.
- In Fig. 4, as in the previous comment 6, please add an appropriate control to understand the alteration.
- In Fig. 5, the green arrow is not explained in the figure legend.
- Lines 185, 186, please indicate in Fig. 6 the area described as “subtle alterations of the outer retinal layers, more pronounced in the left eye”.
Author Response
Reviewer 3
Girbardt et al. report the six cases of retinal vascular events shortly after administration of mRNA or adenoviral vector-based COVID-19 vaccines. The authors raise the questions of a direct correlation of the adverse responses of these cases against the COVID-19 vaccines as possible side effects with potential serious post-immunization complications of COVID-19 vaccinations. The report is timely and would be very important. However, healthy control images were not presented in some figures. And images were not understandable for non-experts without assisting explanations. Below are my specific comments.
- Introduction can cover more about mRNA and adenovirus vector vaccines in terms of efficacy and safety concerns, and how these previous reports could be related to the current report.
Response 1: We thank the reviewer for the helpful comment. We have edited the introduction accordingly (p.2 lines 44-52).
- Safety is often emphasized in mRNA vaccines over viral vectors, but the reported cases in this manuscript appear to show both types of vaccines, mRNA and adenovirus vectors, showed similar levels of safety concerns. It would be beneficial to discuss this point.
Response 2: The reviewer raised a very important point. We edited the discussion accordingly (p.13 lines 283-287).
3.Please add control to Fig. 1 so readers can easily understand the alteration.
- Line 67, missing retinal blood supply in this area. Where does this explain? Adding an arrow to the area would be helpful.
Response to point 3 and 4: The reviewer made a very valid comment on adding normal control images to better understand the pathological changes seen. However, imaging modalities are not always taken in both eyes in clinical routine, especially when no clinical pathologies are seen in the unaffected eye on clinical examination. Thus, in case 1 only the affected eye underwent multimodal imaging. To better understand the shown pathologies, we added a marking in the images and edited the figure legend accordingly. Where control images of the unaffected eye were available, those were added accordingly (for case 3, case 4 and case 5). Changes to the manuscript: Figure 1 + figure legend.
- In Fig. 2, are greed dots the hemorrhages and hard exudates in the left eye?
- In Fig. 2C and D, I understand the vessel leakage, but it is unclear about the description “FFA revealed arterial capillary non-perfusion and a delayed venous filling”.
- In Fig. 2, where were the OCT images taken? And please indicate the areas described as “a hyper-reflectivity of the inner retinal layers in the right eye and a cystoid macular edema”.
Response to point 5-7: We have added white arrows to the images to mark the pathological changes seen. The FFA of the right eye in case 2 shows an arterial capillary non-perfusion as marked by the arrows which is a result of the arterial occlusion. The delayed venous filling can`t be seen in the images but is only noticed by stopping the time by the examiner between dye administration and observed filling of the retinal veins. A delayed filling also results from a vascular occlusion process. Changes to the manuscript: Figure 2 + figure legend.
- In Fig. 3, please indicate the peri-papillary cotton wool spot. Since these images were taken by different imaging modalities, adding control images would be helpful to understand.
Response 8: We have edited Figure 3 accordingly. Changes to the manuscript: Figure 3 + figure legend.
- In Fig. 4, as in the previous comment 6, please add an appropriate control to understand the alteration.
Response 9: We have added control images of the healthy fellow eye of the patient. Changes to the manuscript: Figure 4 + figure legend.
- In Fig. 5, the green arrow is not explained in the figure legend.
Response 10: The figure legend has been edited accordingly. Furthermore, we have edited control images of the unaffected eye. Changes to the manuscript: Figure 5 + figure legend.
- Lines 185, 186, please indicate in Fig. 6 the area described as “subtle alterations of the outer retinal layers, more pronounced in the left eye”.
Response 11: White arrows were added to the images to highlight the pathological changes and explained in the figure legend. Changes to the manuscript: Figure 6 + figure legend.

Round 2
Reviewer 1 Report
The authors have addressed all my review comments, now it is suitable for publication in "Vaccines" journal.
Thanks for the invitation!